# Experiences of community reintegration after obstetric fistula repair at Jean Paul 2 hospital, Conakry, Guinea

Hady Kaba[1]*, Momo Aboubacar Touré[2], Mandian Camara[3], Mira Johri[1,4]

1 École de Santé Publique de l'Université de Montréal - Département de Gestion, Évaluation et Politique de Santé, Montréal, Canada, 2 JHPIEGO -Unité de Coordination Régionale du Projet HSD, Kindia, Guinée, 3 Hôpital Jean Paul 2 - Service de Gynécologie-Obstétrique, Conakry, Guinée, 4 CRCHUM - Centre de Recherche du Centre Hospitalier de l'Université de Montréal, Montréal, Canada

* Hady.kaba@umontreal.ca

**Data Availability Statement:** Our research data is stored in the Dataverse of the University of Montreal: Kaba, H. (2024). Experiences of community reintegration after surgical repair of

## Abstract

The study explored women's experiences of their community reintegration process after surgical repair of obstetric fistula at Jean Paul 2 Hospital in Conakry, Guinea. The study examined how lived experiences of the disease impacted on the community reintegration of treated women. Using a qualitative research methodology, ten women participated after giving informed consent. Semi- structured interviews, lasting an average of 30 to 60 minutes were guided by an interview guide. The main themes covered were experiences with the disease, perceived social support and reintegration into the community. According to the participants, delays in obstetric care were the main cause of obstetric fistula. Socio- economic, cultural and medical factors such as early marriage, lack of education and poverty contributed to these delays. Even after surgical repair, women continued to endure the physical and psychosocial consequences of the disease, exposing them to stigma, discrimination and even rejection within the community. Study participants also reported a lack of social support. The little support perceived by these women concerned food and medical needs. This made the women dependent on their families. Reintegration into the community also proved difficult due to the persistent silence and stigma surrounding their situation. The study results highlight the complexity of the challenges faced by women with obstetric fistula in their journey towards integration. Effective management of obstetric fistula requires a holistic approach victims, their communities, health professionals and decision-makers in solving this problem. So, to improve women's reintegration after treatment for obstetric fistula, it is vital to raise awareness of the causes and consequences of the disease among the women concerned and their families, and to provide rapid access to emergency obstetric care, reinforce social support and set up economic empowerment programs.

## Introduction

Obstetric fistula is a serious condition that devastates the quality of life of millions of women worldwide, especially those living in low-income areas with limited access to adequate obstetric care.

obstetric fistula : Case study at Jean Paul 2 Hospital in Conakry, Guinea. (VERSION PROVISOIRE) [jeu de données]. Borealis. https://doi.org/10.5683/SP3/3WEKGK.

**Funding:** The author(s) received no specific funding for this work.

**Competing interests:** The authors have declared that no competing interests exist.

According to the World Health Organization, more than two million women currently live with obstetric fistula, and between 50,000 and 100,000 women worldwide are affected each year. The scourge mainly affects sub-Saharan Africa and South Asia. In these regions, despite underestimation of the scale of the problem due to the stigmatizing nature of the disease and lack of access to obstetric care, the prevalence of obstetric fistula could be as high as 3 cases per 1,000 live births.

This condition results from dystocic labor during childbirth, leading to abnormal communication between the bladder and vagina, or between the rectum and vagina, due to a mechanical defect in the progression of the fetus through the genitopelvic tract [1,2]. Obstetric fistula is a tragedy that shows the extent to which health systems in developing countries fail to provide quality maternal health care [3]. The lives of women suffering from obstetric fistula remain heavily impacted in terms of physical, economic and psychosocial health [4–6]. This scourge, long neglected by medical research but experienced daily by millions of women around the world in the poorest socio-economic environments, is now attracting renewed interest from the scientific community.

Surgical repair of obstetric fistula, resulting in closure of the abnormal communication, remains a central element of treatment, offering real hope of recovery for women victims. However, full recovery requires more than surgery, it also requires full reintegration of survivors into their communities after surgery.

In Guinea, an increasing number of studies are exploring the problem of obstetric fistula from an epidemiological and therapeutic standpoint, but very few examine the experience of these victims [7–11]. The lived experiences of patients who suffer or have suffered from obstetric fistula remain a largely unexplored terrain. It is within this framework that our study proposes to explore in depth the impacts and challenges of community reintegration for women with obstetric fistula after surgical repair.

The aim of the present study is therefore to explore the experiences of women with obstetric fistula after treatment at Jean Paul 2 Hospital Conakry-Guinea. More specifically, the study aims to understand the impact of obstetric fistula on women's community reintegration following surgical repair. A better understanding of the impact of the disease and the challenges faced by its victims will contribute to the overall strategy to address obstetrical fistula from a holistic, person-centric approach.

## Methodology

### Study design

We conducted a qualitative study to explore the experience of women operated on for obstetric fistula in their journey towards community reintegration. This approach to analysis is deemed appropriate for an in-depth understanding of the challenges faced by women with obstetric fistula in their aspiration to lead a normal life in their socio-cultural environment. The study was conducted at the Jean Paul 2 Hospital in Conakry, from November to December 2021.

The Jean Paul 2 Hospital, located in the commune of Ratoma in Guinea, is a primary and secondary health care establishment placed under the supervision of the Ministry of Social Affairs. This hospital houses a center specializing in the care and rehabilitation of women who have undergone obstetric fistula repairs. The center has a maternity service with a unit dedicated specifically to women victims of fistula, offering them care and appropriate medical monitoring. The medical team is made up of three obstetrician-gynecologists, two surgeons specializing in fistulas, as well as midwives and social workers.

## Recruitment

Given the complexity of the problem, the study used purposive sampling. The study's target population was women with obstetric fistula who had been admitted and treated at the Jean Paul 2 Hospital, and who eventually returned to live in their socio-cultural environment.

To analyze the community integration of women victims of obstetric fistula after treatment at the Jean Paul 2 Hospital, we chose the period 2017–2020, after consultation with the hospital authorities. This period was chosen for the relevance and availability of recent data. Previous years presented limitations in terms of traceability and availability of data, mainly due to the absence of a computerized patient record management system, with information being recorded in handwritten registers, making their management difficult.

Consultation of this admission register enabled us to identify 25 patient files with at least one means of contact, either a telephone number or a home address. A solicitation to participate was made to this group, and we eventually obtained the agreement of 10 patients residing in their original living environment. All patients awaiting surgical treatment or in hospital were excluded from the study.

Participation was voluntary, and participants signed an informed consent form. No remuneration for participants, except for travel and food expenses.

## Collecting data

For data collection, semi-structured face-to-face interviews were conducted at the Jean Paul 2 center, using an interview guide (S1 Text). This interview guide was first submitted to a group of experts working in the field of obstetric fistula. Their feedback enabled us to identify three main themes:

1. **Life experiences:** this theme explores the personal life stories of women with this condition, from onset to treatment.

2. **Perceived social support:** this theme covers all the help received from family, friends and close acquaintances.

3. **Community integration**: this theme describes women's participation in social and cultural activities within the community.

To facilitate analysis and interpretation, notes were also taken during the meetings. For reasons of discretion, all interviews were held in a dedicated area by the administration of the Jean Paul 2 Hospital and lasted between 30 and 60 minutes on average. A general questionnaire was used to collect the socio-demographic characteristics of the participants, as well as their gyneco-obstetrical history (S2 Text). We report in the annex, the socio-demographic questionnaire and the validated interview guide which were used to collect the data for our study.

## Data management and analysis

The socio-demographic characteristics and gyneco-obstetrical history of the participants were described. Analysis of the data corpus began during the field phase and continued throughout the study. The audio recordings were made in the local language (Soussou, Pulhar) and transcribed into French with the help of field notes. To generate results from the corpus of data collected, an inductive thematic analysis was carried out, following the stages of inductive analysis [12–14]. Using Nvivo12 software, we double-coded the material according to the elements relevant to our study questions. The two codings were confronted to discuss any discrepancies in order to establish a consensus. This approach enabled us to verify the relevance and clarity

of 3 main themes and their sub-themes: (i) living with the disease; (ii) social support; and (iii) degree of integration. During the analysis, all data sources (observations, interviews and field notes) were triangulated to ensure validity of the results in accordance with recognized criteria of rigor [15,16].

### Ethics statement

Ethical approval for this study was obtained from two ethics committees, The first is the Comité d'éthique de la recherche en sciences et en santé (CERSES) of the Université de Montréal, withapproval number 2021–1064. The second is the Comité national d'éthique pour la recherche en santé de la Guinée with approval number 95/CNERS/21.

Written informed consent was obtained from all participants.

## Results

### Socio-demographic and obstetric characteristics of study participants

A total of 10 women agreed to participate in our study. At the time of the study, the age of the participants ranged from 23 to 50 years, with an average of 32.4 years, and the most of them lived in rural areas, 7 out of 10 women. All participants had presented with vesicovaginal fistula. The average age at first marriage was 17.56 years, with a range of 13 to 25 years. On average, these women lived with a fistula for 7.20 years, with a maximum of 20 years. In this sample, 9 out of 10 women had no formal education; 7 out of 10 women were still married and residing in the marital home, and among them, 3 women were cohabiting with co-wives. As for the place of delivery, 6 of the 10 women gave birth to their child at home, while only 2 of the 10 received professional assistance during childbirth. In most cases, 8 out of 10, most women have lost their newborn. At the time of the interviews, half of the women were still involved in income-generating activities.

Inductive thematic analysis of the data corpus enabled us to establish 3 main themes: life experience, social support and degree of integration. To preserve the anonymity of participants after informed consent has been obtained, pseudonyms are used for all quotes in the text.

### Theme1: Experiences with illness: Genesis and evolution of the disease through to treatment

According to the accounts gathered from the participants, delays in providing adequate care during obstructive labour combine with socio-economic, cultural and medical factors in the occurrence of obstetric fistula. At the time of fistula onset, the women in this study lived in remote, impoverished areas with no timely access to adequate health care services. Our study population was characterized by early marriage, lack of school education and information, and poverty, which are considered among other things to be central factors in the formation of obstetric fistula [17]. Here are the testimonies of two participants illustrating these conditions:

> *By the way, I was married to this man at a very young age, even though my breasts hadn't really formed. . .and very soon I had my children. . .and then this illness came along. . .*
>
> *(Angeline)*
>
> *We live in a village far from the hospital and when a health problem arises, it becomes very complicated, because there's no one to help you and you don't have fast transport to get to the hospital*
>
> *(Léa).*

In terms of physical health, participants report an inability to hold urine, as well as unpleasant odors and leg pain, as factors affecting their well-being. As testified by the participants:

*I was no longer able to hold back the urine and it would leak without me realizing it as soon as I stood up. I would have to put on layers of cloth to avoid wetting my clothes. I was constantly washing these diapers several times a day because of the bad odors*

*(Rose).*

*This situation lasted, with the smell of urine becoming increasingly embarrassing for myself and those around me. I was the object of stares from difficult people*

*(Lucy)*

These physical consequences contributed significantly to the deterioration of their psychological and emotional health. Participants in the study often experienced feelings of shame and guilt because of their physical condition. This transformed their lives into an object of discrimination and rejection within the community. Here are some excerpts from participants' testimonies:

*Today I really regret this illness, in the name of God. . . We're happy with our lives when we're in good health. Then one day, you find yourself with a burden you couldn't have imagined, even in your dreams. The people around you call you a witch because you're the only one suffering from this illness. . . The people you know won't want their children to come near you or touch you anymore*

*(Angeline).*

Surgical repair of the obstetric fistula was a decisive factor for these women. In the words of the study's participants, the treatment has brought about major changes in their lives and improved their prospects, as one participant's account testifies.

*It was finally after my operation at the Jean Paul 2 hospital that I was able to have my health. I no longer leak urine and that has really given meaning to my life, because at least I can go to bed without the bed being wet the next day. It's a real relief*

*(Lucy)*

In economic and relational terms, the women in the study live in disadvantaged and remote socio-economic environments. They depend on their spouse's contribution to the household. And in these communities, when faced with the realities of the wife's physical illness, the spouse chooses to abandon the victim to her suffering and remarry. This contributes to the increasing precariousness of these women.

The results of our study reveal that most of these women are kept within their household, sometimes sharing their home with a co-wife. This situation makes life extremely difficult for the victims, both financially and psychologically. Half of the women in the study tried to maintain a source of income by doing small businesses or farm work. However, these efforts proved insufficient to cover all their financial needs.

The lingering consequences of the stigma experienced have also made interpersonal and community interactions difficult for most participants, forcing them to live apart from community members. As participant

*I couldn't say anything without it turning into reproaches and reprimands. My husband was never concerned about my health. I have remained in his home, however, but I assure you that life is not as it was before. Nothing I can say matters*

***(Lucy)***

## Theme2: Experiences of perceived social support: Family, friends, or close acquaintances

Support for obstetric fistula victims is an essential step in their healing process and in restoring their dignity as women in the community. This support could come from family, friends, or close associates. Most participants in the study did not receive the help and support they needed for their physical, mental, and economic health. The little support they experienced was mainly focused on dietary and medication needs. Very often, participants described the indifference of their spouses to their situation and their ability to deal with the stigma attached to them. However, some participants were able to count on the support and backing of family members and friends alike. The following quotes illustrate the experiences of these participants:

*I've been sick all the time and so has my child. But that's none of her business, and I have to fend for myself. So, you can imagine the difficulties I endure, as I have no activity that can give me the necessary means to provide for our needs. My baby and I lack the means for everything we need, even medicine*

(**Lucy**).

*In my own family, my sister is the person I can count on most. Since I've had this disease, she's been the one who helps me out with food and medicine.*

***(Léa).***

## Theme3: Experiences of community integration: Place and role of participants in the home and community

Integrating into the community was a complex challenge for our participants, particularly those who had been abandoned or divorced. This difficulty was exacerbated by the fact that most of these women were heavily dependent on their partners to meet their most basic needs, thus creating a bond of financial and emotional dependence.

In our study, it appeared that, even after undergoing surgical repairs to correct their obstetric fistula, these participants chose to remain silent about their condition. Feelings of shame and low self-esteem played a significant role in keeping their condition quiet, despite their physical recovery.

The social position of these women as members of the household and community was marginalized due to their health conditions, which prevented them from actively participating in community activities. However, it is important to note that there are both enabling factors and obstacles to their reintegration. These factors, whether linked to context, individual personality, or family context, significantly influence their journey. This is how participants highlighted the crucial role of surgical repair in transforming their perspectives, although the psychosocial

and physical after-effects of the disease remained. Most of the women treated expressed their deep gratitude to the healthcare team at Jean Paul 2 hospital for their support, advice, and accompaniment during this difficult period.

However, upon returning to their usual environment, most participants continued to remain silent about their condition, even after completing the surgical repair. Social support may have helped some of them regain their place as women within the community, but unfortunately, for the majority, this support proved almost non-existent, thus constituting an obstacle to their full reintegration. Participant Nathalie reported the following extract:

*Relationships with people in the village are good, since nobody knew what had happened to me. I can take part in social activities to the best of my ability without being looked at or judged by people, since nobody knows what I've been through with this disease.*

*(Nathalie)*

In addition, other socio-cultural and economic factors hampered the return and community reintegration of the participants. In their social relationships, these women were confronted with stigmatization and exclusion. Community activities and ceremonies are at the heart of social functioning in these poor, remote regions. For the women, any loss of control over urination represents a very painful event, contrary to widespread social norms and expectations in the community, leading to feelings of shame, guilt and, above all, loss of social status as a woman. These feelings were widely reported by the participants in our study, forcing victims to isolate themselves even further from the rest of the community. A woman without a social role feels useless to the rest of her community.

Quotes from two participants illustrate these experiences:

*I can't explain this situation to everyone because it makes me feel ashamed, because people can't understand that it's an illness and I'd be judged, besides, because of the smells around me I know they're already talking about it behind my back.*

*(Rose)*.

*I don't take part in any ceremonies, and my older sister refuses to let me, knowing the difficulties I've been through with this illness.*

*(Bernadette)*.

## Discussion

In our study, we sought to understand how the experience of women with obstetric fistula affects their community reintegration after surgical treatment. Focusing on three themes, namely experiences with the disease, perceived social support, and community integration, our findings provided an in-depth look at the reintegration challenges faced by women who underwent surgical repair of obstetric fistula at the Jean Paul 2 Hospital in Conakry, Guinea. To provide a more holistic understanding of the problem, our results were compared with the light of other studies carried out in different regions, particularly in sub- Saharan Africa. This approach enabled us to identify trends and similarities, as well as implications for future interventions.

Regarding their experiences with the disease, the participants experienced delays in their initial care due to socio-economic, cultural, and medical factors. Early marriage, lack of

education, poverty and geographical remoteness were central factors in the occurrence of obstetric fistula. This finding is consistent with previous research in Uganda [18], Burkina Faso [19] and Ethiopia [20] who also identified these factors as key determinants in the occurrence of obstetric fistula. In our study, the physical consequences of obstetric fistula, such as urinary incontinence, physical pain, and unpleasant odours, had a negative impact on the physical and psychosocial health of the women affected. Even after successful surgery to repair obstetric fistula, these after-effects contributed to their stigmatization, discrimination and sometimes exclusion within their communities. Similar experiences have been reported in other regions affected by fistula [21].

In terms of perceived social support, most participants did not receive the necessary support from family, friends, or close associates. What little support they did receive was mainly related to food and medication needs, while spouses were often indifferent to their situation. This finding is in line with previous research which also identified a lack of adequate social support in similar contexts.

The process of community integration was a difficult one for most of the women in the study. Even without being abandoned or divorced, these women faced a reality where their status with their spouse was diminished, finding themselves replaced by a co-wife. In this case, they live with their husbands out of social responsibility, but no longer have the same place as before. This highlights the fragility of women's social and economic status in society, a problem which does not seem to affect men in the same way. Exclusion from community activities and silence about their state of health are manifestations of the persistent stigmatization endured by these women. While surgery was instrumental in eliminating physical complications such as incontinence, the challenges of reintegration remain. These challenges are linked, on the one hand, to economic and relational problems and, on the other, to the lasting after-effects of stigmatization and social exclusion. This conclusion is in line with previous studies that have also highlighted the maintenance of stigma and social exclusion after surgical repair [21].

It is important to note that, despite similarities with other studies, countries' geographical contexts may present specificities likely to influence the many challenges faced by women operated on for obstetric fistula in their community reintegration process. This is why we would like to remind you of the importance of the disparities in the different health care systems and support programs offered to obstetric fistula victims, depending on the geographical regions concerned [22]. These disparities may act as an element influencing the quality of post-operative care and the availability of the support needed to enable successful community reintegration for operated women.

Thus, our study, while sharing similarities with previous research, broadens our understanding of the challenges of reintegrating women who have survived obstetric fistula in the context of Guinea. By integrating the lessons learned from these studies into the existing scientific literature, we can formulate more precise recommendations for improving maternal health policies and obstetric fistula management programs, tailored to the contextual and specific needs of Guinea.

## Strengths and limitations

The results of our study add to our existing knowledge of the life experiences of women with obstetric fistula in a difficult socio-economic environment. The choice of qualitative analysis is explained by our desire to gain an in-depth understanding of the participants' experiences. For a population of 10 participants, we do not pretend to generalize to all victims of obstetric fistula in the Guinean context, as we may have failed to reach certain women for whom the challenges

were even greater in their process of returning home. This constitutes a limitation of our study, which was conducted during the Covid-19 pandemic and confronted with the reality of field accessibility, financial and time constraints.

## Conclusion

In-depth exploration of the challenges associated with community reintegration after surgical repair of obstetric fistula was at the heart of our study conducted at the Jean Paul 2 Hospital in Conakry, Guinea. The obstacles women face, both medical and psychosocial, during their community reintegration process, remain complex and influenced by several factors.

The main findings of our research underline the fact that women suffering from obstetric fistula face delays in treatment due to socio-economic, cultural, and medical factors. These factors include early marriage, lack of education, poverty, and geographical remoteness.

Despite the persistent after-effects of stigmatization and discrimination in the community, surgical repair of fistula remains an essential step towards improving the physical and psychosocial health of women victims.

Also, the study participants were confronted with a lack of social, financial, and emotional support. They had to rely mainly on limited support from family or friends.

Finally, community reintegration proved to be a challenge for the participants, due to the silence surrounding their condition and persistent stigmatization. For this reason, a more holistic approach to the care of women with obstetric fistula needs to be advocated. This must involve victims, health professionals, policymakers, and community members alike, to overcome this problem.

## Recommendations

The recommendations arising from the findings of our study on the challenges of community reintegration after surgical repair of obstetric fistula at the Jean Paul 2 Hospital in Conakry, Guinea, reveal the need for increased efforts to raise awareness of the issue within communities in Guinea. These efforts must focus on understanding the causes and consequences of obstetric fistula through reproductive health education programs.

Efforts must therefore be focused on providing women with rapid access to emergency obstetric care, especially in the most remote and disadvantaged regions, thus helping to reduce delays.

In terms of social support, greater awareness is needed among families and spouses of the importance of emotional and financial support for women who have undergone fistula repair surgery. Psychosocial support programs need to be set up to help these women cope with persistent stigmatization.

Finally, the community reintegration of women treated for obstetric fistula must be facilitated by the development of programs designed to boost their self-confidence and help them regain their place in society. This important aspect must include economic empowerment programs, such as vocational training and access to micro-credits to develop income-generating activities.

## Supporting information

**S1 Checklist.**
(DOCX)

**S1 Text. Interview guide.**
(DOCX)

**S2 Text. Medical and obstetrical history.**
(DOCX)

## Author Contributions

**Conceptualization:** Hady Kaba, Momo Aboubacar Touré, Mandian Camara, Mira Johri.

**Data curation:** Hady Kaba.

**Investigation:** Hady Kaba.

**Methodology:** Hady Kaba.

**Project administration:** Hady Kaba.

**Writing – original draft:** Hady Kaba.

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
