## [Editor Report · Decision Letter 0]

1 Mar 2024

PGPH-D-24-00111

Experiences of community reintegration after surgical repair of obstetric fistula: Case study at Jean Paul 2 Hospital in Conakry, Guinea.

Dear Dr. Kaba,

Thank you for submitting your manuscript to PLOS Global Public Health. After careful consideration, we feel that it has merit but does not fully meet PLOS Global Public Health’s publication criteria as it currently stands. Therefore, we invite you to submit a revised version of the manuscript that addresses the points raised during the review process.

1. Please note we recommend that authors use the use the COREQ checklist, or other relevant checklists listed by the Equator Network, such as the SRQR, to ensure complete reporting.(http://journals.plos.org/globalpublichealth/s/submission-guidelines#loc-qualitative-research). Please include this information in the methods section and amend content to ensure criteria is met.

2. Please clarify in the text if names included with quotes are the names of participants or pseudonyms and if REB consent was obtained for this type of reporting.

3. Currently, results are reported in the Methods section. Would recommend moving the results section for clarity.

4. The title of this study indicates that it is a “Case Study”, which is not justified in the methods section. Please consider changing the paper's title or justifying the case study methodology.

5. Please complete all items on the Human Participants Research Checklist that are relevant for your submission by following this link: http://journals.plos.org/plosone/s/file?id=dc11/PLOSOne_Human_Subjects_Research_Checklist.docx.

There may be overlap between the checklist items and other queries listed here; please address any duplicated queries both in your response email and on the checklist itself. Upload the completed Human Participants Research checklist as file type “Other” when you re-submit your manuscript. This document is for internal journal use only and will not be published if your article is accepted. The requested information will help us to assess whether your submission complies with PLOS’s policies and adheres to applicable reporting standards. Note that your manuscript may be rejected if you provide incomplete or inadequate responses to the checklist questions and that changing your article's ‘Section/Category’ does not affect this requirement.

We look forward to receiving your revised manuscript.

Kind regards,

Archna Gupta, MD, CCFP, MPH, PhD

Academic Editor
---

## [Decision Letter · Decision Letter 1]

21 Jun 2024

PGPH-D-24-00111R1

Experiences of community reintegration after surgical repair of obstetric fistula: Case study at Jean Paul 2 Hospital in Conakry, Guinea.

Dear Dr. Kaba,

Thank you for submitting your manuscript to PLOS Global Public Health. After careful consideration, we feel that it has merit but does not fully meet PLOS Global Public Health’s publication criteria as it currently stands. Therefore, we invite you to submit a revised version of the manuscript that addresses the points raised during the review process.

The manuscript has been reviewed by a second reviewer and their comments are available below. The reviewer has made suggestions to provide additional information to improve the clarity of the results presented as well as the context. 

<ul><li> 

A rebuttal letter that responds to each point raised by the editor and reviewer(s). You should upload this letter as a separate file labeled 'Response to Reviewers'.<li> 

A marked-up copy of your manuscript that highlights changes made to the original version. You should upload this as a separate file labeled 'Revised Manuscript with Track Changes'.<li> 

We look forward to receiving your revised manuscript.

Kind regards,

Archna Gupta, MD, CCFP, MPH, PhD

Academic Editor

Journal Requirements:

Additional Editor Comments (if provided):

Reviewers' comments:

Reviewer's Responses to Questions

**Comments to the Author**

1. If the authors have adequately addressed your comments raised in a previous round of review and you feel that this manuscript is now acceptable for publication, you may indicate that here to bypass the “Comments to the Author” section, enter your conflict of interest statement in the “Confidential to Editor” section, and submit your "Accept" recommendation.

Reviewer #1: All comments have been addressed

2. Does this manuscript meet PLOS Global Public Health’s publication criteria? Is the manuscript technically sound, and do the data support the conclusions? The manuscript must describe methodologically and ethically rigorous research with conclusions that are appropriately drawn based on the data presented.

Reviewer #1: Yes

3. Has the statistical analysis been performed appropriately and rigorously?

Reviewer #1: N/A

4. Have the authors made all data underlying the findings in their manuscript fully available (please refer to the Data Availability Statement at the start of the manuscript PDF file)?

Reviewer #1: Yes

5. Is the manuscript presented in an intelligible fashion and written in standard English?

Reviewer #1: Yes

6. Review Comments to the Author

Reviewer #1: The authors made a good attempt to share insights into the challenges faced by women who had obstetric fistulas in Conakry, Guinea and environs.

Here are my comments :

The authors should add the validated questionnaire sample for review. 

Correct spelling error, i.e., recommendation, not recommandation. 

Why the choice of 2017–2020 considering the low incidence of obstetric fistulas in the hospital, as evidenced by 25 cases in three years?

Some information about this hospital might also be beneficial. Is it a primary, secondary, or tertiary health facility, which may play a role in patient volume and the complexity of cases managed?

7. PLOS authors have the option to publish the peer review history of their article (what does this mean?). If published, this will include your full peer review and any attached files.

**Do you want your identity to be public for this peer review?** For information about this choice, including consent withdrawal, please see our Privacy Policy.

Reviewer #1: **Yes: **Gbolahan Olatunji

---

## [Editor Report · Decision Letter 2]

18 Jul 2024

Expériences de réinsertion communautaire après réparation d'une fistule obstétricale à l'hôpital Jean Paul 2, Conakry, Guinée.

PGPH-D-24-00111R2

Dear Dr Kaba,

We are pleased to inform you that your manuscript 'Expériences de réinsertion communautaire après réparation d'une fistule obstétricale à l'hôpital Jean Paul 2, Conakry, Guinée.' has been provisionally accepted for publication in PLOS Global Public Health.

Best regards,

Archna Gupta, MD, CCFP, MPH, PhD

Academic Editor